# Multispecies Fresh Water Algae Production for Fish Farming Using Rabbit Manure

**Adandé Richard [1,\*], Liady Mouhamadou Nourou Dine [1], Djidohokpin Gildas [1], Adjahouinou Dogbè Clément [1], Azon Mahuan Tobias Césaire [1], Micha Jean-Claude [2] and Fiogbe Didier Emile [1]**

[1] Laboratory of Research on Wetlands (LRZH), Department of Zoology, Faculty of Sciences and Technology, University of Abomey-Calavi, B.P. 526 Cotonou, Benin; liadynouroudine@gmail.com (L.M.N.D.); gdjidohokpin@gmail.com (D.G.); adjaclem@gmail.com (A.D.C.); azonmahunan@yahoo.fr (A.M.T.C.); edfiogbe@yahoo.fr (F.D.E.)

[2] Department of Biology, Research Unit in Environmental Biology, University of Namur, 5000 Namur, Belgium; jean-claude.micha@unamur.be

\* Correspondence: richard_adande@yahoo.fr; Tel.: +229-95583595

**Abstract:** The current study aims at determining the optimal usage conditions of rabbit manure in a multispecies fresh water algae production for fish farming. This purpose, the experimental design is made of six treatments in triplicate including one control $T_0$, $T_1$, $T_2$, $T_3$, $T_4$, $T_5$ corresponding respectively to 0, 300, 600, 900, 1200, 1500 g/m$^3$ of dry rabbit manure put into buckets containing 40 L of demineralized water and then fertilized. The initial average seeding density is made of $4 \times 10^3 \pm 2.5 \times 10^2$ cells/L of Chlorophyceae, $1.5 \times 10^3 \pm 1 \times 10^2$ cells/L of Coscinodiscophyceae, $3 \times 10^3 \pm 1.2 \times 10^2$ cells/L of Conjugatophyceae, $2.8 \times 10^3 \pm 1.5 \times 10^2$ cells/L of Bascillariophyceae, and $2.5 \times 10^3 \pm 1.4 \times 10^2$ cells/L of Euglenophyceae. During the experiments, the effects of these treatments on abiotic and biotic parameters (chlorophyll-a concentration, phytoplankton density and algal density) of different production media were monitored. Results show that average density of different phytoplankton classes is higher in treatment $T_5$ ($7.91 \times 10^8 \pm 6.78 \times 10^7$ cells/L) followed by $T_4$ ($5.56 \times 10^8 \pm 4.27 \times 10^7$ cells/L), $T_2$ ($3.87 \times 10^8 \pm 3.10 \times 10^8$ cells/L), $T_3$ ($3.79 \times 10^8 \pm 3.18 \times 10^8$ cells/L, with high significant difference (F $_{(4,84)}$ = 5, 35, $p < 0.00$). Chl-a concentration varied from $0.07 \pm 0.05$ mg/L ($T_0$) to $14.47 \pm 12.50$ mg/L ($T_5$) with high significant differences observed among treatments (F $_{(5,83)}$ = 3,09, $p = 0,01$). In addition, fourteen (14) species belonging to eight (8) families, five (5) classes and three (3) phyla were identified in our different production media. During the culture, Chlorophyceae class was the most represented in all treatments with 5 species (36% of the specific diversity) while Euglenophyceae class (7%) was the least represented with only one (01) species. According to these results, treatments $T_2$ (600 g/m$^3$), $T_3$ (900 g/m$^3$) and $T_4$ (1200 g/m$^3$) of dry rabbit manure are those worthy to be recommended as an alternative for a low cost massive production of multispecies freshwater algae that can be easily used by freshwater zooplankton and macroinvertebrates. Indeed, despite the best performances that it shows, treatment $T_5$, presents important eutrophication's risks.

**Keywords:** aquaculture; multispecies algae; rabbit manure; production; zooplankton

## 1. Introduction

Animal wastes are valorized in aquaculture as fertilizers to increase halieutic production in order to meet the crescent fish demand observed these latter years [1–4]. Due to their low cost and low pollution risks when recycling and treatments cautions are taken off stream of production system,

certain animal wastes such as rabbit manure whose mineral salts richness is well known are used at the detriment of chemical fertilizers to increase primary production in fish farming systems in order to produce fresh water zooplankton and benthic macro-invertebrate that constitute the main food source for fish larvae and fries [5–9] In some western countries, modern technologies are used to produce important algae biomass for fish farming from these wastes [10], unfortunately these technologies are often highly expensive and not adapted to developing countries. In developing countries where sun light and temperature are naturally high to favor the utilization of mineral salts (ammonium, nitrate, nitrite and orthophosphate) by algae, simple and low cost algal production technics from animal wastes can be used in fish farming [4,8,9,11]. Algae are food for zooplankton and benthic macro-invertebrates that must be rich in nutrients (lipids, poly-unsaturated fatty acids, polysaccharides, carotenoids, steroids, and premium vitamins for fish growth [12] in order to guarantee a good fish production. However, very few studies in developing countries are focused on the improvement of plankton culture [1,13] though judicious use of various sources of animal genuine organic fertilizers could enable important production of algae at low cost to feed zooplankton and benthic macro-invertebrates in order to optimize profitability in rural fish farming [9,14]. The nutritional quality of plankton is influenced by fertilization and the water quality and determines the composition of phytoplankton species in culture media [1,15], and is the reason why the aim of this study is to determine the optimal dose of rabbit manure for multi-species production of fresh water algae for fish farming.

## 2. Material and Methods

### 2.1. Experimental Design

The experimental design comprises 18 plastic buckets of 80 L capacity (previously cleaned and disinfected), exposed to the opened air at the Research Station for Fish Farming Diversification of the University of Abomey-Calavi (UAC). These buckets are filled with 40 L of demineralized water immediately fertilized with manure of rabbit fed on cheap price food diet made of 2% cassava, 30% corn bran, 10% palmist cake, 10% soya cake, 5% cotton cake, 2% shell, 10% malt, 5% beer yeast, 10% *Panicum maximum*, and 1% of salt [14]. Six (6) treatments were tested in triplicate: $T_0$ (the control), $T_1$, $T_2$, $T_3$, $T_4$, $T_5$ corresponding respectively to 0, 300, 600, 900, 1200, 1500 g of dry rabbit manure (RM) in one (1) $m^3$ of water. These manures are made of 15.02% of Nitrogen (N), 1.26% of phosphorus (P) and 0.84% of potassium (K) with ratio N/P = 11.89.

### 2.2. Seeding, Identification and Counting of Phytoplankton

At the beginning of the experiment, production media (water) containing rabbit manure was previously abandoned for three days to enable fertilization by nutrients contained in rabbit manure, and then they were seeded with phytoplankton. Seeding has been achieved by sampling 10 L of water from multi-culture (*Clarias gariepinus* and *Oreochromis niloticus*) pond and gently filtering it according to Agadjihouèdé et al. [16] by using a 25-µm plankton net in order to eliminate zooplankton, then the obtained filtrate was added to culture medium in each bucket for phytoplankton seeding. A fresh part has been used for observation and identification of phytoplankton species with photonic binocular microscope (BI 100; VWR International Belgium) and another 100 mL part was treated with formaldehyde 5% and concentrated 100 times for phytoplankton counting using a Neubauer counting cell under a photonic microscope [17]. Identification was carried out according to Adjahouinou et al. [17] from photographs realized at 10×, 40× and, 100× according to height and mobility of algae species and by using the identification keys of Bourrelly [18–20], Compère [21–26], Iltis [27], and Guiry et al. [28]. Algal density was estimated according to Boauli et al. [29] through the equation:

D (cells/L) = $M \times \frac{C}{V} \times 10^5$ with M: mean algae cell number per rectangle of hematimeter, C: concentration coefficient of the ampoule, V: sample volume. Thus, the initial mean seeding density was $4 \times 10^3 \pm 2.5 \times 10^2$ cells/L of Chlorophyceae, $1.5 \times 10^3 \pm 1 \times 10^2$ cells/L of Coscinodiscophyceae,

$3 \times 10^3 \pm 1.2 \times 10^2$ cells/L of Conjugatophyceae, $2.8 \times 10^3 \pm 1.5 \times 10^2$ cells/L of Bacillariophyceae and $2.5 \times 10^3 \pm 1.4 \times 10^2$ cells/L of Euglenophyceae.

During experiment monitoring, for each phytoplankton sampling, the culture medium was homogenized, then 5 L of water were filtered with a 25-μm plankton net and concentrated 100 times.

### 2.3. Monitoring of Physico-Chemical and Trophic Parameters

During the experiment, temperature (°C), pH, conductivity (μs/cm), and dissolved oxygen (mg/L) were measured in situ every three days at 12 PM with a CALYPSO, Champigny-Marne, France multi-parameter device, sensitivity ±0.1 °C (Version Soft/2015, SN-ODEOA 2138). At each sampling, 1/2 L of the culture medium was filtered for determination of chlorophyll-a (Chl-a) content, algal biomass, and nutritive dissolved salts concentrations (ammoniac nitrogen, nitrate, nitrite, and orthophosphates). The dosage of Chl-a was carried out according to the protocol described in AFNOR NF T90-117 standards. Nutritive salts were determined according to Rodier et al. [30] using a HACH DR/2800 molecular absorption spectrophotometer.

### 2.4. Statistical Analyses

Statistical analyses were carried out with STATISTICA software (Statsoft inc., Tulsa, OK, USA) at a threshold of 5%. A general linear model (GLM) was used followed by the Bonferroni test to compare the means of pytoplankton densities. The effect on different physicochemical parameters were analyzed using repeated measures ANOVA after checking the sphericity conditions. The values reported in the results are means and standard deviations. The correlation between the concentration of chl-a and the treatments was analyzed using the Pearson correlation test.

## 3. Results

### 3.1. Abiotic Parameters

Except temperature, mean values of physico-chemical parameters varied from one treatment to another from 22.12 to 50.63 for conductivity, from 0.72 to 1.05 for pH, from 0.72 to 2.95 for dissolved oxygen, from 0.01 to 0.03 for salinity, from 8.47 to 27.11 for TDS, and from 0.37 to 1.51 for transparence. As illustrated in Table 1, mean values of pH, electrical conductivity, salinity, and TDS were higher in fertilized media than in control treatment $T_0$ with high significant difference ($p < 0.00$). Except pH, the highest mean values of different parameters prior quoted were recorded in medium $T_5$. The same remark was made for dissolved oxygen though concentration was most important in treatments $T_2$ and $T_5$, followed by $T_4$ and $T_3$ with significant difference ($p < 0.05$). Transparence decreased progressively from control medium $T_0$ to the most fertilized medium $T_5$.

**Table 1.** Physico-chemical parameters of the different treatments.

| Parameters | $T_0$ | $T_1$ | $T_2$ | $T_3$ | $T_4$ | $T_5$ | F |
|---|---|---|---|---|---|---|---|
| T°C | 33.74 ± 1.70 | 34.02 ± 1.81 | 34.30 ± 1.70 | 34.34 ± 1.87 | 34.48 ± 1.86 | 34.51 ± 1.78 | $F_{(5, 12)}$ =12.63 |
| Cond (μs/cm) | 37.66 ± 23.17 c | 597.82 ± 34.10 b | 604.50 ± 23.17 b | 607.19 ± 22.12 b | 625.58 ± 36.08 ab | 664.45 ± 50.63 a | $F_{(5,12)}$ = 41.86 |
| pH | 6.87 ± 0.94 b | 7.29 ± 0.79 a | 7.33 ± 0.94 a | 7.12 ± 0.92 a | 7.09 ± 1.05 a | 7.16 ± 1.05 a | $F_{(5,12)}$ = 5.53 |
| DO (mg/L) | 3.81 ± 0.72 d | 7.48 ± 1.95 c | 9.42 ± 0.72 a | 8.58 ± 2.40 a | 8.99 ± 2.95 a | 9.66 ± 3.33 a | $F_{(5, 12)}$ = 59.40 |
| Sal (mg/L) | 0.08 ± 0.01 c | 0.31 ± 0.02 b | 0.32 ± 0.01 b | 0.32 ± 0.01 b | 0.33 ± 0.01 ab | 0.36 ± 0.03 a | $F_{(5, 12)}$ = 9.33 |
| TDS | 64.60 ± 8.77 c | 296.92 ± 17.07 b | 302.56 ± 8.77 b | 303.43 ± 7.47 b | 318.76 ± 16.81 ab | 338.61 ± 27.11 a | $F_{(5, 12)}$ = 7.88 |
| Transp (cm) | 27.91 ± 0.37 a | 22.50 ± 1.13 b | 16.85 ± 1.51 ce | 17.63 ± 1.41 de | 14.43 ± 1.38 fh | 12.98 ± 1.50 gh | $F_{(5, 12)}$ = 17.69 |

Cond: conductivity, DO: dissolved oxygen, Sal: salinity, Transp: transparence, TDS: total dissolved solids. $T_0$ (the control), $T_1$, $T_2$, $T_3$, $T_4$, $T_5$ corresponding respectively to 0, 300, 600, 900, 1200, 1500 g of dry rabbit manure (RM). Mean values affected by different letters on the same line are significantly different at 5% threshold. Mean concentrations of nutrients varied significantly ($p < 0.05$) from one treatment to another (Figure 1). The lowest mean values ($0.06 \pm 0.12$ mg/L; $0.07 \pm 0.11$ mg/L; $0.08 \pm 0.10$ mg/L and $0.5 \pm 0.13$ mg/L respectively for $N-NO_3^-$, $N-NO_2^-$, $N-NH_3$, and $P-PO_4^{3-}$) were obtained in the control treatment ($T_0$) while the highest ($2.82 \pm 0.75$ mg/L; $0.75 \pm 0.35$ mg/L; $0.81 \pm 0.14$ mg/L and $3.15 \pm 0.16$ mg/L) were recorded in treatment $T_5$.

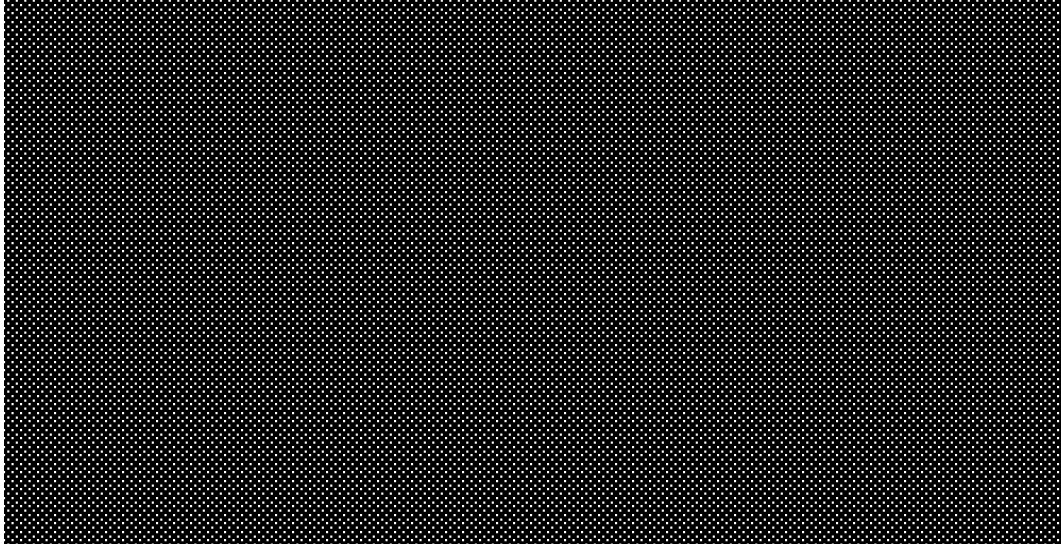

**Figure 1.** Mean concentrations of nutrients in the different culture media. For the same parameters, means followed by the same letters are not significantly different ($p > 0.05$). $T_0$ (the control), $T_1$, $T_2$, $T_3$, $T_4$, and $T_5$ correspond, respectively, to 0, 300, 600, 900, 1200, 1500 g of dry rabbit manure (RM).

### 3.2. Biotic Parameters

### 3.2.1. Chlorophyll-a Concentrations

Figure 2 shows that mean concentrations of chl-a obtained during the experiment varied from $0.07 \pm 0.05$ mg/L ($T_0$) to $14.47 \pm 12.50$ mg/L ($T_5$) with high significant difference observed among treatments ($F_{(5,12)} = 9554.9$, $p < 0.00$). In the same way, high positive correlation ($R = 0.88$ $p < 0.00$) was observed between chlorophyll-a concentration and rabbit manure dose.

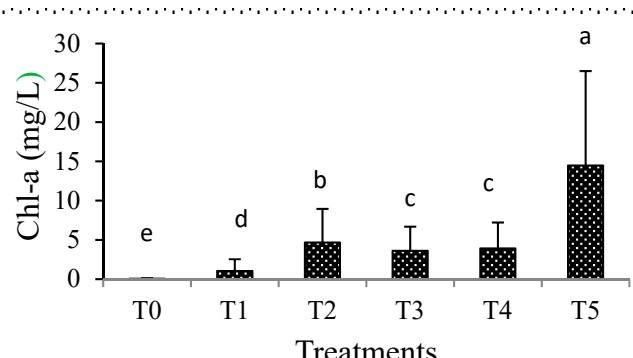

**Figure 2.** Mean concentrations of chlorophyll-a in different treatments: $T_0$ (the control), $T_1$, $T_2$, $T_3$, $T_4$, and $T_5$ correspond, respectively, to 0, 300, 600, 900, 1200, 1500 g of dry rabbit manure (RM).

### 3.2.2. Phytoplankton Diversity

A total of fourteen (14) phytoplankton species belonging to eight (8) families, five (5) classes, and three (3) phyla were identified in the production media (Table 2).

**Table 2.** Phytoplankton taxons identified and produced from rabbit manure.

| Phylum | Classes | Families | Genuses and Species |
|---|---|---|---|
| Bacillariophytes | Coscinodiscophyceae | Melosiraceae | *Melosira italica* |
| | | | *Melosira sp* |
| | Bacillariophyceae | Pinnulariaceae | *Pinnularia viridis* |
| | | | *Pinnularia sp* |
| | | Naviculaceae | *Navicula sp* |
| Charophytes | Conjugatophyceae | Closteriaceae | *Closterium sp* |
| | | Desmidiaceae | *Staurastrum margaritaceum* |
| | | | *Staurastrum pinnatum* |
| | Chlorophyceae | Scenedesmaceae | *Coelastrum sp* |
| | | | *Scenedesmus javanensis* |
| | | | *Scenedesmus apiculatus* |
| | | | *Scenedesmus quadricauda* |
| | | Palmellopsidaceae | *Asterococcus sp* |
| Euglenophytes | Euglenophyceae | Euglenaceae | *Euglena ehrenbergii* |

### 3.2.3. Phytoplankton Density

The evolution of phytoplankton density varies according to phytoplankton classes and applied treatments (Figures 3 and 4). During the production period, two to three peaks were observed according to the treatments. An increase in the algal density of different classes was observed from 3rd in all treatments with a first peak at the 3rd day in treatment $T_1$ and the 6th day for treatments $T_0$, $T_2$, $T_3$, $T_4$, and $T_5$ (Figure 3). During the peak, the highest densities were those of Chlorophyceae and Bacillariophyceae, respectively, in treatments $T_4$ ($5.14 \times 10^8 \pm 1.69 \times 10^8$ cells/L) and $T_5$ ($8.60 \times 10^8 \pm 1.19 \times 10^8$ cell/L). From the 9th day, algae density decreased in all media but growth started enabling a second peak at the 12th day, although with lower densities than the first. From the 15th day till the end of the experiment, phytoplankton density decreased in treatments $T_3$, $T_4$, and $T_5$. However, a third peak was observed at the 21st day in treatments $T_1$ and $T_2$ with lower densities than the second and at the 24th day in treatment $T_0$ with almost the same density as the second. On the contrary, the mean density obtained in the first and second peak in which Chlorophyceae and Bacillariophyceae were predominant, that of the third peak is dominated by Conjugatophyceae, Chlorophyceae, and Bacillariophyceae (Figures 3 and 4), respectively, in treatments $T_0$ ($3.33 \times 10^5 \pm 2.21 \times 10^5$ cells/L), $T_1$ ($2.40 \times 10^7 \pm 1.10 \times 10^6$ cells/L), and $T_2$ ($2.47 \times 10^7 \pm 1.67 \times 10^6$ cells/L). The mean total density for the whole phytoplankton classes during experiment period was higher in treatment $T_5$ ($7.91 \times 10^8 \pm 6.78 \times 10^7$ cells/L) followed by $T_4$ ($5.56 \times 10^8 \pm 4.27 \times 10^7$ cells/L), $T_2$ ($3.87 \times 10^8 \pm 3.10 \times 10^8$ cells/L), $T_3$ ($3.79 \times 10^8 \pm 3.18 \times 10^8$ cells/L, $T_1$ ($2.28 \times 10^8 \pm 1.49 \times 10^8$ cells/L), and $T_0$ ($4.5 \times 10^7 \pm 6.06 \times 10^6$ cells/L) with high significant difference ($F_{(4, 84)} = 5.35$, $p < 0.00$). Mean densities obtained in treatments $T_2$, $T_1$, and $T_3$ were not significantly different among them. In return, we observe significant difference between treatment $T_0$ and others in the one hand and among treatments $T_4$, $T_5$, and $T_2$, and $T_1$ on the other hand ($p < 0.05$). Globally, in terms of total density, treatment can be classified as follows: $T_5 > T_4 > T_2 > T_3 > T_1 > T_0$.

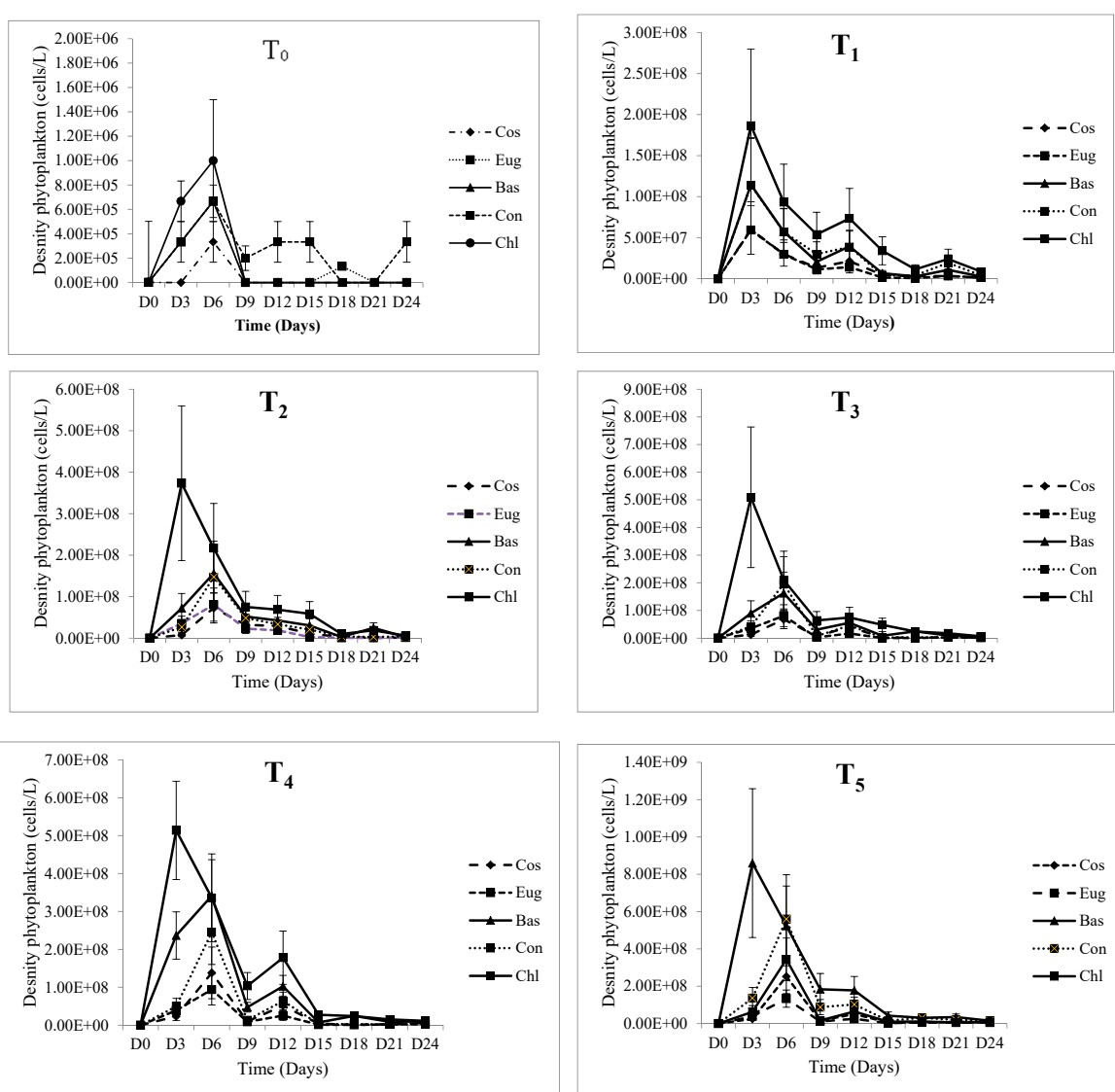

**Figure 3.** Phytoplankton density evolution in the classes per treatment (dose of rabbit manure). Cos: Coscinodiscophyceae, Eug: Euglenophyceae, Bac: Bacillariophyceae, Con: Conjugatophyceae, Chl: Chlorophyceae. T0 (the control), $T_1$, $T_2$, $T_3$, $T_4$, $T_5$ corresponding respectively to 0, 300, 600, 900, 1200, 1500 g of dry rabbit manure (RM).

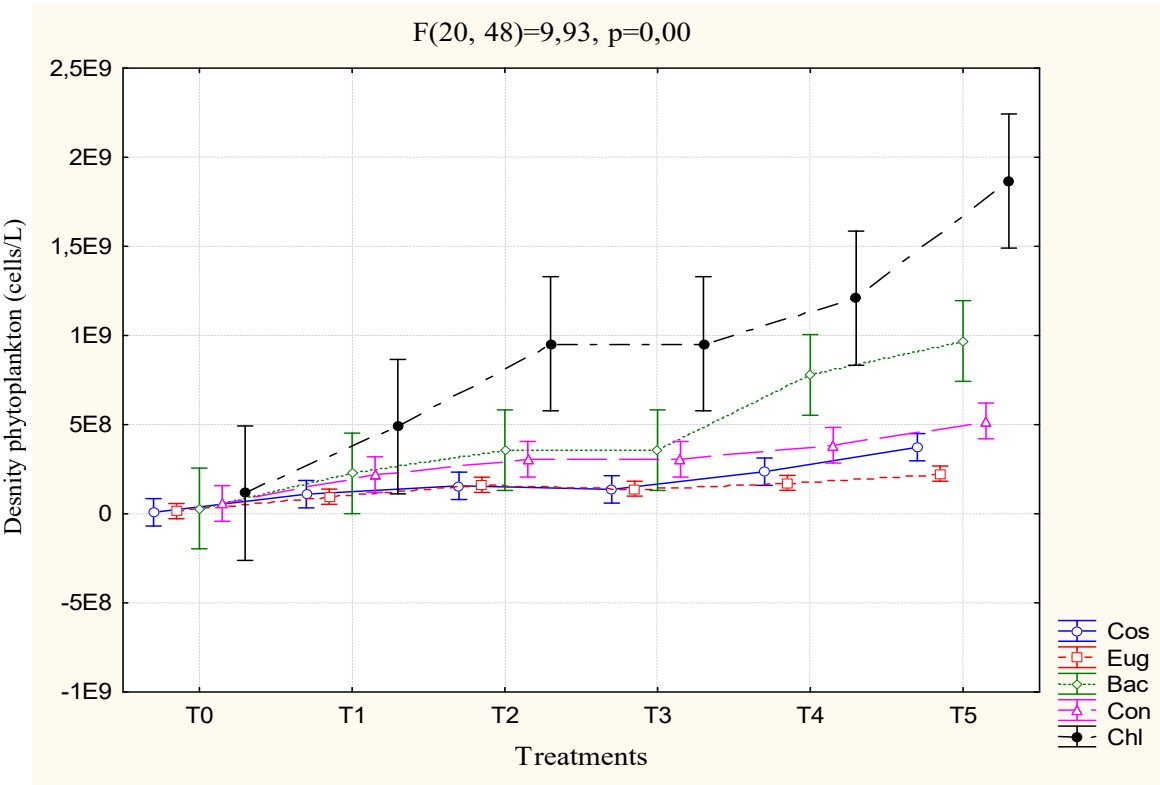

**Figure 4.** Evolution of different phytoplankton phylum. T: treatments Cos: Coscinodiscophyceae, Eug: Euglenophyceae, Bac: Bacillariophyceae, Con: Conjugatophyceae, Chl: Chlorophyceae. T0 (the control), T1, T2, T3, T4, T5 corresponding respectively to 0, 300, 600, 900, 1200, 1500 g of dry rabbit manure (RM).

## 4. Discussion

Several studies focused on freshwater and marine phytoplankton showed the importance of environmental conditions [31–33]. Indeed, the mean temperature of 34 °C recorded in our culture media was higher than 25 °C and near 35 °C obtained respectively during the production of *Isochrysis galbana* [31] and *Scenedesmus abundans* [34]. According to the same authors, at these temperatures, algae are able to synthetize maximum quantity of glucides and lipids contrary to proteins that are optimally synthesized at lower temperature (15 °C). At this temperature, sun light allow algae to consume the $CO_2$ released through bacterial mineralization of organic matter and to produce oxygen that is indispensable to aquatic living beings, as well as bacteria that use it in mineralization processes and ammonium oxidation [35,36]. These facts justify low ammonium rates and high nitrate concentrations observed in the current study (Figure 1). Thus, the main methods to eliminate the ammonium that come from organic matter (rabbit manure) could be volatilization and nitrification [29,37]. The pH value recorded in culture media varied from 6.85 to 7.16 and is near that recorded by Hodaifa et al. [38], which is 7 during the production of *Scenedesmus obliquus*. According to Hodaifa et al. [38], neutral pH media are favorable to maximal proteins and chlorophylls synthesis by algae. In return, pH values recorded indicate low photosynthetic activity of algae; this could be explained by the fact that it could be accumulated at the ampoule of buckets due to the absence of mixing and would not benefit from the available light. These observations are in accordance with those of Zimmo et al. [39] according to which, in waste water ponds, photosynthesis of algae was limited by the water column and then low pH values comprised 6.9–7.3 were observed. Concerning dissolved oxygen, the best concentrations were recorded in fertilized media (7–9 mg $O_2$/L). Indeed, these high dissolved oxygen concentrations obtained in our culture media through algae photosynthesis [40] average 34 °C confirm

the link existing between algae photosynthetic activity and temperature. These results corroborate with those of Bouali et al. [29] who obtained the highest oxygen rate (8.15 mg $O_2$/L) during the dry season. In addition, this author proves the existence of high correlation between the dry period and the increase of oxygen level. High electrical conductivities of fertilized media indicate high mineralization and testify their mineral salts richness [4,30]. It results from this analysis that the rabbit manure used has highly affected the water quality in the different freshwater algae culture media. Orthophosphate concentration from 0.06 to 2.82 mg P-$PO_4^{3-}$/L assimilable by algae recorded in culture media is near that observed by Toyub et al. [41] ranging from 1.5 to 2.5 mg P-$PO_4^3$/L orthophosphates for which production of *Scenedesmus obliquus* is maximal. Temperature and nutrient concentrations (nitrogen and phosphorus) constitute key factors to modify the predominance of the different phytoplankton species during the culture period. The highest density obtained in Chlorophyceae could be due to the different concentrations of inorganic nitrogen in culture media that range from 0.8 to 4 mg N/L recommended by Boyd [42] and Schlumberger et al. [43] for good growth of this phytoplankton class. High orthophosphate concentrations (0.5–3.5 mg/L) recorded were higher than that recommended (0.2–0.5 mg/L of orthophosphate) by the same authors in fish ponds dominated by Chlorophyceae beyond which the trend could be in favor of other species with more affinity. The high transparence value of Secchi disk in treatment $T_0$ contrary to the other fertilized treatments associated to the high orthophosphate concentration and pH inferior to 9 could forecast predominance of Chlorophyceae [43]. The highest of this latter was recorded in treatment $T_5$ and could be the source of low density of Chlorophyceae to the profit of Bacillariophyceae (Coscinodiscophyceae). Barbe et al. [44] reported that ammonium and orthophosphates concentrations higher than those recommended as well as ratio P-$PO_4$/Ni (inorganic nitrogen) superior to 1/8 and 1/10 could provoke development of Cyanobacteria even we notice their absence in our culture media. Indeed, this high phosphorus concentration could be due to the death of certain algae and bacteria in the culture medium and could constitute the second source of organic fertilizer. In addition, the highest phytoplankton biomass obtained in treatment $T_5$ is tied to its phosphorus richness. Indeed, phosphorus constitutes limiting factor to phytoplankton production in aquatic medium [43]. However, the most assimilable fraction is orthophosphate [45] and often represents a small part of free phosphate in culture media [11,46,47].

Additionally, the highest algae density ($7.91 \times 10^8 \pm 6.78 \times 10^7$ cells/L) and the highest mean rate of chlorophyll-a (14.47 mg/L) recorded in treatment $T_5$ ($1500 g/m^3$) could be tied to trophic enrichment phenomenon in fertilized media that could lead to high organic load. Consequently, the use of treatment $T_5$ could provoke eutrophication in fish production systems. Indeed, 14 mg/L of chlorophyll-a is the maximal value recommended by Agadjihouèdé et al. [11]. It would be judicious not to use this dose in fish farming ponds to avoid excessive development of certain toxic and indigestible Cyanobacteria. However, the high Cyanobacteria density known by its nutritional richness and digestibility, constitute a trump for optimization of zooplankton and micro-invertebrates. Mean chlorophyll-a concentrations obtained in treatments $T_2$ (4.67mg/L), $T_3$ (3.60 mg/L) and $T_4$ (3.92 mg/L) were superior to the minimal value of 2 mg/L proposed by Canovas et al. [48] for good primary production. Thus, $T_2$, $T_3$, and $T_4$ would offer very good nutritive conditions to algae. Algae densities obtained in $T_2$, $T_3$, and $T_4$ were higher than those obtained by Toyub et al. [41] that are, respectively, 97.05; 83.21; 65.19, and 51.21 ($\times 10^2$ cells/L) cultivated from soft drink factory effluents at different concentrations. By the same way, mean densities obtained in these studies are higher than those obtained by Sipaúba-Tavares et al. [1] during the culture of *Ankistrodesmus gracilis* ($144 \times 10^1$ cells/L) based on NPK. These differences could be explained by the nature of the substrates and culture media.

In addition, the analysis of phytoplankton classes in our culture media sets first Chlorophyceae, followed by Bacillariophyceae, Conjugatophyceae, Coscinodiscophyceae, and Euglenophyceae, which constitute predominant algae groups in fish farming ponds revealing nutrient availability. However, low densities of Euglenophyceae could be due to low organic load [17,27,29,49,50] but also consumption by zooplankton. Additionally, Chlorophyceae and Euglenophyceae are easily edible by many zooplankton, such as Cladocera, belonging to the *Daphnia* genus, as well as small

zooplankton, such as rotifers [43,51]. According to Barbe et al. [44] and Schlumberger et al. [43], Bacillariophyceae (Diatomeae) and Conjugatophyceae (Desmidials) are considered as secondary species according to nutrient concentrations in production medium and also present in oligotrophic water. This approach of algae production is being improved to generate high algae biomass that constitutes primary production essential not only for fish production but also for renewable energies [4,52,53]. Analysis of physico-chemical parameters, in general, and especially dissolved oxygen, conductivity, pH, and nutrients concentration (nitrogen and phosphorus) as well as the different densities of phytoplankton classes showed that genuine animal organic fertilizers (rabbit manure) are suitable for primary production favoring optimal fish farming at a cheap price.

## 5. Conclusions

The current study shows the valorization of rabbit manure through the production of multispecies freshwater algae for fish farming. It results from this study that treatment $T_2$, $T_3$, $T_4$, and $T_5$, respectively 600, 900, 1200, and 1500 ($g/m^3$), provided good algae productions, while, treatment $T_5$ shows a risk of eutrophication. In application, treatments $T_2$, $T_3$, and $T_4$ could constitute rabbit manure doses to be used for massive production of multi-species freshwater phytoplankton easily edible by zooplankton and macro-invertebrates. Beyond the current quantitative evaluation, it is important to envisage in forthcoming studies the bromatological analysis of these algae in order to estimate their nutritive components.

**Author Contributions:** Conceptualization, A.R.; methodology, L.M.N.D.; software, D.G.; validation, A.D.C. and A.M.T.C.; formal analysis, M.J.-C.; investigation, M.J.-C.; resources, F.D.E. All authors have read and agreed to the published version of the manuscript.

**Funding:** This research received no external funding.

**Acknowledgments:** We thank the Ministry of Higher Education and Scientific Research (MESRS) of Benin Republic that funded this work through the project "Training of Trainers of Benin Universities".

**Conflicts of Interest:** The authors declare no conflict of interest.

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
