# Peer review of "Multispecies Fresh Water Algae Production for Fish Farming Using Rabbit Manure"

_fishes, doi:10.3390/fishes5040035_

Round 1
Reviewer 1 Report
Please find my comments below:
- Line 64: could you add the name of some salts which are used to increase the primary production in fish farming system for zooplankton?
- Line 96: could you add information about the added salt?
- Figure 1: could you correct the information about chart legend?
- Figure 4: in how many repetitions each experiment was performed. If at least three repetitions were made, please add the standard deviation in the diagrams.
- Line 291: Please explain the equation to me. I think it contains errors. Ni = NH4 + N-NO3- + N-NO2-
Author Response
thank you.

Reviewer 2 Report
The paper entitled “Multispecific production of freshwater algae based on rabbit manure: Growth, density, systematic and quality of water for fish farming purpose” deals with an experimental study aimed at investigating the suitability of using rabbit manure as feed for algae cultivation. Although the study presents interesting results, the paper needs to be improved in the use of language (I am not an English speaker, therefore I am not much confident in giving suggestions in English, although some mistakes are highlighted in the specific comments section of this review) and in presentation and discussion of results by enhancing the quality of graphs and extending the discussion section.
Specific comments
Line 69, Check the correctness of references’ annotation;
Line 71-74. The following sentences: “In certain occidental countries, modern technologies use these animal wastes for important algae biomass production in fish farming [13]. These different production technologies of occidental countries are too expensive and often non-adapted to developing countries” contain elements of vagueness, i.e, “certain occidental”…and “these different…”, Please enrich the meaning of the sentences by removing elements of vagueness;
Line 96. number in brackets can be removed;
Line 104. See comment for line 96
Lines 107 and 109. Replace “bottom” with a more specific term;
Line 110. See comment for line 69;
Line 112. See comment for line 96;
Lines 116-117. Report the units of terms of the analytic expression;
Lines 118-120. Here the authors use the unit cells/L, whereas in the abstract ind/L. please make consistent the text;
Line 126. Check the correctness of the verb “serve”;
Line 137. As the abbreviation of chlorophyll-a, was already used before, use here the abbreviation Chl-a;
Line 143. Table 1. Reduce the size of characters in caption;
Line 158. Figure 1. Correct P-PO43- in the legend and set the minimum of Y-axis at 0. Reduce the characters size in the caption;
Line 167. See comment for line 137;
Line 179. See comment for line 137;
Line 175. Figure 2. Reduce the size of characters in caption;
Lines 179-180. See comment for line 96;
Line 182. See comment for line 96;
Figure 3 and Table 2. See comment for Figure 2;
Figure 4. Homogenize the size of characters used in different graphs;
Figure 4. See comment for figure2;
Line 221. Replace “picks” with peaks;
Lines 223,224, 226, 228, 230 and 231. See comment for line 221;
Figure 5. See comment for Figure 2;
Line 265. What is the meaning of “an.”?
Line 298. Check the correctness of “raison”;
Line 300. Replace “Ni” with N;
Line 307. See comment for line 96;
Line 308. See comment for line 137;
Author Response
suggestion taken into account. please see the file

Reviewer 3 Report
The authors present data that are of possible interest of a wide audience. They should improve the presentation of the data and have a clear structure of the paper. Clearly stating research questions and hypotheses are necessary steps to improve the paper and that would benefit also the flow of the discussion. I have some suggestions on the data analysis and presentation that would benefit if integrated.
Point-by-point suggestions:
- The abstract should have the same structure as the paper: introduction, research question, methods, results, discussion, conclusion. At the moment the results are 80% of the abstract.
- Line 64. You should expand on macronutrients and their importance. Also why you report N and P only? What about K?
- Check in-text reference order, it is often not correct (e.g. line 75).
- Line 77. Which method? You did not introduce any method that you want to consider in your study.
- Line 88. Missing research questions and predictions.
- In materials and methods, please use the active voice.
- Line 99. Do you mean the macronutrient composition of the manure is? How did you reach those percentages of NPK? It would be important to show how you prepared the manure. Also why your manure is so high in N?
- Line 103. How was it observed? Did you do tests? Did you base your choice on published papers?
- Line 104. change "ten (10)" to 10L. Same in other parts.
- Lines 118-120. How did you come out with these calculations?
- Statistical analysis. This should be called repeated-measures ANOVA. A Linear Mixed Model would be a better approach for your data as it would be much more powerful. RM ANOVAs are better for simpler designs, while linear mixed models allow for a more flexible approach. I have to say RM ANOVAs are not incorrect in your case (unless some of the variables are not normally distributed). By looking at the variance of some of your variables, I am not sure the assumption of normality is met for all of them. The other assumption to test is the sphericity if you want to use RM ANOVAs. The best approach would be using Generalised Linear Mixed Models so you can fit your dependent variables with different fit functions. That would work to test the difference in the density of cells as well, as I guess it is an integer and can be fit to Poisson distributions. There is also a problem using LSD post-hoc test, and that is that it is not correcting for multiple comparisons. You should use some correction (e.g. Bonferroni). That is valid also if you change the analyses to GLMMs.
- Results in general. What type of variability over the mean do you report, I assume it is standard deviation but it should be specified.
- Line 155. Transparence does not seem to decrease progressively as T3 is either higher or the same as T2, and looking at the variance there is not much difference between T4 and T5
- In the graphs, it is not clear the use of letters. Also, what are error bars? Standard deviations?
- Lines 169-171. I am not sure why you use a correlation. It would make much more sense to use a GLMM.
- Figures 3 and 5 can be summarised in a few lines in the results. No need for figures with so few information.
- Table 2. It is weird to have photos of some species and not others. This table could be much smaller if you remove the photos.
- Figure 4. It is difficult to compare between experiments as the Y-axis has different scales. It would be much more powerful to keep the same scale. Also, why this graph does not have variance over mean? It would be better to use the mean per bucket and test as you do for the other parameters.
- Discussion. You should discuss your main findings first, following your research questions and predictions. The discussion should be more structured, now it is in a single paragraph
Author Response

(The authors gave the same response as above.)

Round 2
Reviewer 2 Report
The authors have improved the paper but not enough to be published in the current version. Therefore, I invite them to revise again the English language (still they use pick in place of peak) and overall to use graphs with axis labels in English and not in French and moreover to write the text in accordance with graphs...for example T0 in the graph is J0
Reviewer 3 Report
It is necessary that the authors reply more extensively than "according to your suggestions" to the reviewer's comments as for most of them they did not take any action. In particular, regarding statistics, need to specify that you met the assumptions to run RM ANOVAs (Is it true? Which tests you used to test assumptions?) and why you use LSD post-hoc tests (incorrect) instead of having a corrected post-hoc test. This is the main concern, but there are other comments I made that were not taken into account. It might be ok if you did not take into account all of my previous comments, but I expect some explanation on why you did not integrate those comments. And please clarify how you integrated the suggestions if you integrated them. That is a necessary step before considering this new revision. So please reconsider the previous revision i made and clearly specify what you changed and why you did not consider some of the changes suggested even if you then replied "according to your suggestions". In addition, the new version presents figures in French; they must be changed.
Round 3
Reviewer 3 Report
I have only a few minor issues:
- when reporting the exponential numbers, you cannot report them like 5.14.108, you need to report them like 5.14 x 108 or 5.14 x 10E8.
- Write somewhere in the data analysis section in the methods that the values reported in the results are means and standard deviations
- line 142-143. I was saying that transparence is not really decreasing progressively as T2 and T3 are the same and there is probably no statistical difference between T4 and T5 (I am very confident that 14.43 +- 1.41 is not statistically different from 12.98 +-1.50). So the wording should be changed.
